# Health Effects of Resveratrol: Results from Human Intervention Trials

**DOI:** 10.3390/nu10121892

**Published:** 2018-12-03

**Authors:** Sonia L. Ramírez-Garza, Emily P. Laveriano-Santos, María Marhuenda-Muñoz, Carolina E. Storniolo, Anna Tresserra-Rimbau, Anna Vallverdú-Queralt, Rosa M. Lamuela-Raventós

**Affiliations:** 1Department of Nutrition, Food Science and Gastronomy, School of Pharmacy and Food Sciences XaRTA, Institute of Nutrition and Food Safety (INSA-UB), University of Barcelona, 08921 Santa Coloma de Gramenet, Spain; sonialrmz@gmail.com (S.L.R.-G.); elaversa21@alumnes.ub.edu (E.P.L.-S.); mmarhuendam@ub.edu (M.M.-M.); carolina.e.storniolo@gmail.com (C.E.S.); avallverdu@ub.edu (A.V.-Q.); 2CIBER Physiopathology of Obesity and Nutrition (CIBEROBN), Institute of Health Carlos III, 28029 Madrid, Spain; 3Human Nutrition Unit, University Hospital of Sant Joan de Reus, Department of Biochemistry and Biotechnology, Faculty of Medicine and Health Sciences, Pere Virgili Health Research Center, University Rovira i Virgili, 43201 Reus, Tarragona, Spain; anna.tresserra@iispv.cat

**Keywords:** bioavailability, antioxidant, obesity, metabolic diseases

## Abstract

The effect of resveratrol (RV) intake has been reviewed in several studies performed in humans with different health status. The purpose of this review is to summarize the results of clinical trials of the last decade, in which RV was determined in biological samples such as human plasma, urine, and feces. The topics covered include RV bioavailability, pharmacokinetics, effects on cardiovascular diseases, cognitive diseases, cancer, type 2 diabetes (T2D), oxidative stress, and inflammation states. The overview of the recent research reveals a clear tendency to identify RV in plasma, showing that its supplementation is safe. Furthermore, RV bioavailability depends on several factors such as dose, associated food matrix, or time of ingestion. Notably, enterohepatic recirculation of RV has been observed, and RV is largely excreted in the urine within the first four hours after consumption. Much of the research on RV in the last 10 years has focused on its effects on pathologies related to oxidative stress, inflammatory biomarkers, T2D, cardiovascular diseases, and neurological diseases.

## 1. Introduction

In the last decades, lifestyle changes, especially in dietary patterns, have been increasingly seen as a means of preventing and treating chronic diseases. In this context, polyphenols have emerged as natural compounds with wide-ranging beneficial effects against cardiovascular diseases (CVD) and cancer [1,2]. Polyphenols are metabolized by the intestine, hepatic cells, or intestinal microbiota. The intestinal absorption, bioavailability, and pharmacokinetics of polyphenols are conditioned by the food matrix in which they are ingested [3,4].

Resveratrol (RV), a naturally occurring polyphenol (*trans*-3,4′,5-trihydroxystilbene), possesses anti-inflammatory, anti-tumorigenic, and antioxidant properties, which may be harnessed in strategies against chronic diseases. The main sources of RV include, above all, grapes (*Vitis vinifera* L.), a variety of berries, peanuts, medicinal plants such as Japanese knotweed [5], and red wine.

Studies on the health benefits of RV have reported a reduction in age-associated symptoms and the prevention of early mortality in obese animals [6,7,8]. The life expectancy of some small organisms has been extended by RV via the stimulation of caloric restriction [9] and the delay of specific age-related phenotypes, e.g., abnormal glucose metabolism [10]. RV is also associated with a slowing down or prevention of cognitive deterioration [11].

The potential mechanisms of action responsible for the health effects of RV are numerous [5]. As RV triggers the expression of a wide range of antioxidant enzymes, determining the contribution of each mechanism to an overall decrease in oxidative stress is a complex task [12]. Additionally, a large number of receptors, kinases, and other enzymes interact with RV, which may influence its biological effects.

The activities of sirtuin 1 (SIRT1) and adenosine monophosphate-activated protein kinase (AMPK), enzymatic regulators of metabolism in multiple tissues, are stimulated by RV in vivo [5,13,14]. Some of the beneficial effects of RV are due to the overexpression of SIRT1 [15]. Moreover, RV has been reported to be a potent inhibitor of quinone reductase 2 activity, which is associated with neurological disorders, although more research is required to confirm this hypothesis [16]. The determination of all effects of RV in humans remains a major challenge.

After oral ingestion, RV is metabolized in the liver to glucuronide and sulphate forms and in the intestine by hydrogenation of the aliphatic double bond [17,18]. RV has been found in urine samples of subjects who have drunk a glass of wine per week or three glasses per week three or five days after the last consumption, respectively [19].

The aim of this review is to summarize the health effects of RV in humans as reported by studies carried out in the last decade, including the determination of plasma, urine, and feces RV and its metabolites. The information is presented in two sections: bioavailability and pharmacokinetics of RV and the effects of RV in different health status.

## 2. Bioavailability and Pharmacokinetics of Resveratrol

After oral administration, RV is absorbed by passive diffusion or by forming complexes with membrane transporters followed by release into the bloodstream, where it can be found mainly as a glucuronide, sulfate, or free [20]. Phase II metabolism of RV or metabolites occurs in the liver, after an enterohepatic transport in the bile that may lead to some RV return to the small intestine [1].

Human clinical trials with RV showed its rapid metabolism [20,21]; it occurs in the liver and promotes the production of conjugated glucuronides and sulfate metabolites, which have biological activity [20]. The metabolites identified were measured by high-performance liquid chromatography analysis, followed by mass spectrometry.

Different clinical trials have found that the majority of plasma RV metabolites are RV-3-*O*-sulfate, RV-4′-*O*-glucuronide, and RV-3-*O*-glucuronide. RV-3-*O*-sulfate circulating levels showed the highest peak concentration compared to the other conjugated RV metabolites [22,23,24,25], except in one study where RV-3-*O*-glucuronide presented the highest peak concentration when the dose of RV was 2.5 g [25]. Pharmacokinetic studies revealed that RV concentration in plasma depends on the doses ingested [22,25,26,27].

In this section, we address research on the bioavailability and pharmacokinetics of RV in human clinical trials during the last 10 years. The subsections are organized according to the effects of different factors.

### 2.1. Effect of Pharmaceutical Formulation and Particle Size

Different strategies have been developed to improve RV efficacy [28], including pharmaceutical manipulation. A novel soluble formulation of *trans*-RV (caplets) was administered to 15 healthy subjects; the same amount of *trans*-RV (single dose of 40 mg) was also administered in dry powder (capsules). There were significant differences in bioavailability between the formulations, being the Cmax (maximum concentration) in plasma 8.8-fold higher for the soluble formulation [29].

RV can be absorbed, metabolized, and excreted in urine, in which up to 21 metabolites of RV have been identified [30]. The intake of 4.7 mg of RV in grape extract tablets (as a nutraceutical) resulted in a delay of the urinary excretion of RV metabolites up to 4-fold higher compared with the intake of 6.3 mg of RV in red wine [31]. This indicates a delayed absorption of RV when it is ingested in grape extract tablets. As a consequence, RV stays longer in the gut and could be metabolized by the gut microbiota. Therefore, RV supplementation can be a good source of this polyphenol.

Furthermore, reducing the size of the chemical particles can increase their absorption and kinetics [32]. The daily intake of 5 g microparticulate RV with a particle size of less than 5 mm (SRT501) for 14 days produced a higher peak plasma concentration than the equivalent dose of non-micronized RV [33], therefore a small particle size improves RV bioavailability.

### 2.2. Matrix Effect

The food matrix is important for RV bioavailability and pharmakocinetics. The bioavailability of RV was assessed when it was consumed together with food, quercetin, or ethanol in a study where 2000 mg of *trans*-RV were administered twice daily for seven days to eight healthy subjects. In this clinical trial, the combined intake with other phenols such as quercetin and alcohol did not influence *trans*-RV pharmacokinetics. However, when RV was ingested within a high-fat breakfast compared with a standard breakfast, the area under the plasma concentration–time curve and the maximum plasma concentration were lower (45% and 46% decrease, respectively) [34]. In a two-way crossover study, 24 healthy subjects were administered 400 mg of *trans*-RV with a high-fat content meal or in fasting conditions [35], and it was concluded that the presence of high-fat food delayed the rate of absorption of *trans*-RV but not the extent of absorption.

The solubility of *trans*-RV in water containing dextrose, fructose, ribose, sucrose, or xylitol was analyzed and compared. The best result was obtained with the ribose solution. A mixture of ribose and 146 ± 5.5 mg *trans*-RV was administered to two healthy human participants, leading to a higher and quicker RV release than that obtained with traditional free RV capsules [36]. A similar result was demonstrated when 250 mg *trans*-RV doses were administered with 20 mg of piperine to 23 healthy adults [23]. In this context, it is possible that the affinity for or the solubility of *trans*-RV in the presence of different substances such as soluble formulations, ribose, and piperine improves RV bioavailability when compared to the classic formulations (capsules).

Furthermore, a study with 36 healthy males ingesting capsules containing 800 mg polyphenols with protein-rich dairy, soy, fruit-flavored drinks, or water, showed that the intake of polyphenols incorporated in protein-rich drinks did not change significantly the bioavailability of polyphenols or their metabolites [37].

A similar result was obtained when 59 high-risk adult subjects at high cardiovascular risk drank 272 mL of red wine (RW, 30 g ethanol/day) or dealcoholized red wine (DRW) every day for four weeks. The RV effect was independent of the alcohol in the red wine [38].

### 2.3. Effects of Other Factors

Repeated oral supplementation of *trans*-RV may influence pharmacokinetic variables. In this context, 13 doses of 25, 50, 100, or 150 mg *trans*-RV were administered six times/day to four groups of eight healthy adults. Differences were observed between the peak *trans*-RV plasma concentrations after ingestion of the 1st and 13th dose, demonstrating the highest peak concentration for the latter. Moreover, *trans*-RV pharmacokinetic values were higher at 8 am and 12 pm [39], meaning that circadian variation and repeated doses affected the bioavailability.

It is important to remember that a wide range of factors such as gut microbiota composition, hormones, gender, and other interindividual differences can modify the structure of RV [40,41]. The gut microbiota plays an important role in the structure of RV, which can affect human health. In feces of healthy humans, *Slackia Equolifaciens* sp. and *Adlercreutzia Equolifaciens* sp. have been identified as dihydroresveratrol producers [40].

### 2.4. Is It Safe to Consume Resveratrol?

Clinical trials have shown that RV and *trans*-RV supplementation is safe and well tolerated at different doses [25,29,33,39,42,43,44,45]. However, some participants reported one or more adverse events, such as gastrointestinal symptoms including nausea, flatulence, bowel motions, abdominal discomfort, loose stools, and diarrhea, after ingesting a dose of 2.5 to 5 g of RV [22,25,34,42,43].

On the other hand, *trans*-RV half-life was one to three hours following single doses and two to five hours following repeated dosing [39]. It is noteworthy that the most important phase of RV excretion occurs in the first four hours after ingestion. Moreover, there exists a relationship between RV levels in plasma and in stools, indicating an enterohepatic recirculation [43].

For the above-mentioned reasons, the bioavailability and pharmacokinetics of RV depend on the doses ingested, the ingestion of food matrix, the particle size, and the role of the gut microbiota in the metabolism of RV. Last, RV intake is safe at a dose of up to 5 g; however, adverse reactions have been observed at higher doses, which should be considered in future studies.

Figure 1 recapitulates the bioavailability and pharmacokinetics of RV. The image schematizes the results of different studies in which RV was identified in urine and blood. Additionally, Table 1 presents the details of the bioavailability and pharmacokinetics of RV in each study considered; it is organized according to health status and year of publication.

## 3. Different Health Effects of Resveratrol

This section addresses research on the effect of RV in human clinical trials in the last 10 years. The subsections are organized according to the different health status or diseases under research.

### 3.1. Effects of Resveratrol on Neurological Diseases and Cognitive Performance

RV is associated with a slowing down or prevention of cognitive deterioration [11]. Although few clinical trials have focused on the effect of RV on Alzheimer’s disease (AD), two studies suggest that RV may change some AD biomarkers. Both studies are rated class II because more than two primary outcomes were designated. A dose of 500 mg RV/day was administered to patients with mild to moderate AD, with 500 mg increments every 13 weeks up to 52 weeks, ending with 1000 mg twice daily. In one of the studies, the brain volume decreased in the RV group; however, the mechanisms of this event were unclear, and cognitive deterioration was not indicated. Both the RV and placebo group showed a decline of Aβ40 (beta amyloid) levels in the cerebrospinal fluid (CSF) or plasma at 52 weeks [24]. A subsequent study reported similar results for CSF Aβ40 compared to baseline, whereas a greater reduction of CSF Aβ42 occurred in the placebo group when compared to the group receiving the RV treatment, indicating that RV could attenuate the progressive decline of this biomarker of AD. In plasma, RV increased MMP10 (matrix metalloproteinase) and reduced IL-12P40 (interleukin) and IL-12P70. Compared to the placebo, the RV treatment reduced MMP9 in CSF [47]. From the evidence described above, RV may regulate neuro-inflammation in AD patients; however, more studies are needed to draw conclusions about RV efficacy in AD.

In 36 adult patients with type 2 diabetes (T2D), a single dose of 75 mg of RV ingested at weekly intervals showed significant changes, enhancing neurovascular coupling capacity and improving cognitive performance [27].

On the other hand, single doses of 250 and 500 mg *trans*-RV in two different days, in healthy subjects, improved cerebral blood flow variables, with increases of total hemoglobin (Hb) and oxygenated Hb (oxy-Hb) and a reduction of deoxy-Hb concentration; nevertheless, it did not produce any change in cognitive performance variables [48]. In a consecutive study, a similar result was reported with 23 healthy adults who ingested two capsules with a dose of 250 mg *trans*-RV and 20 mg of piperine (a pepper-derived alkaloid) in three different days. A significant increase in total-Hb and oxy-Hb was observed, but without any improvement in cognitive functioning [23].

A possible explanation for the contrasting results could be that a different methodology to evaluate cognitive performance was used in these studies. When a more objective measure is used, such as near-IR spectroscopy, the results are not as clear as when the results are based on cognitive test batteries. Likewise, it could be interesting to know the capsule composition, because it could contribute to the observed different effect.

The effects of RV have been observed in other neurological diseases such as Friedreich ataxia. The effect of 5 g of RV ingested daily for 12 weeks was studied in patients diagnosed with the aforementioned disease, showing an improvement in neurologic function, audiologic and speech measures, and oxidative stress marker plasma F2-isoprostane [25].

### 3.2. Effects of Resveratrol on Diabetes Mellitus

RV enhances the endothelial function, increases liver fatty acid oxidation, and decreases oxidative stress [49], leading to an improvement in insulin sensitivity [50]. A study in which 10 overweight individuals with impaired glucose tolerance were administered 1, 1.5, or 2 g of RV per day for four weeks, showed that insulin sensitivity and postprandial glucose levels were improved by RV intake [45]. In another clinical study, 17 volunteers with T2D were treated with 150 mg/day of RV for 30 days, after which, intrahepatic lipid content and systolic blood pressure decreased. Similarly, when overweight and obese men were administered RV for two weeks, 1 g in the first week and 2 g in the second, a reduction in intestinal and hepatic lipoprotein particle production was observed. RV diminished the production rate of ApoB-48 and both the production and fractional catabolic rates of ApoB-100, compared to a placebo [51].

Nevertheless, RV did not enhance hepatic and peripheral insulin sensitivity, which could be explained by a negative interaction with metformin in the patients receiving this kind of treatment [52]. Similar results were reported in 20 overweight or obese men with non-alcoholic fatty liver disease when given a daily dose of 3 g RV for eight weeks. No reduction in insulin resistance, steatosis, abdominal fat distribution, plasma lipids, or antioxidant activity was observed. However, an increase of alanine and aspartate aminotransferases with RV supplementation was observed, due to RV increased hepatic stress [42].

Some studies have analyzed the effect of RV on obesity. A daily dose of 150 mg/day of RV for four weeks was given to 10 slightly obese adults. RV supplementation suppressed postprandial glucagon, which may be important for the treatment of T2D because an excess of this hormone contributes to patient hyperglycemia [53]. Contrasting results can be explained by the effect of the pharmacokinetics of different medications on RV or by liver’s health, because both RV and other medications could be metabolized in the liver; nevertheless, the main reason could be the dose taken.

### 3.3. Effect of Resveratrol on Cancer

The insulin-like growth factor (IGF) signaling pathway, including IGFs, IGF-binding proteins (IGFBP), and IGF receptors, is related to the anticarcinogenic effects linked to dietary restriction. In parallel, RV can act as a chemopreventive agent and a calorie-restriction mimetic in humans [9,54]. Related to these effects, RV decreased IGF-I and IGFBP-3 in 40 healthy volunteers who consumed RV at 0.5, 1.0, 2.5, or 5.0 g daily for 29 days, leading to a reduction of cancer risk. The highest reduction was observed with a 2.5 g dose. Therefore, it was concluded that the IGF system could act as a biomarker of RV chemopreventive action in humans [22].

### 3.4. Effect of Resveratrol on Cardiovascular Diseases

Flow-mediated dilatation (FMD) of the brachial artery is a biomarker of endothelial function and cardiovascular health, with notable importance as an indicator of structural and functional endothelium changes [55,56]. A 270 mg dose of RV administered weekly for four weeks to 14 overweight or obese men or five post-menopausal women with untreated borderline hypertension significantly increased FMD [26]. In another study, RV in red wine was associated with improved levels of glucose and triglycerides, as well as a lower heart rate, whereas no effect was observed for total cholesterol, HDL, LDL, and high blood pressure [57]. However, in overweight individuals, the intake of 150 mg per day of RV for four weeks did not influence metabolic risk markers such as endothelial function or inflammation, which are related to cardiovascular health risk [58,59]. A possible reason for the contrasting results could be related to the ingested dose of RV, because a positive effect of RV in higher doses on other diseases has been observed. However, it is important to note that the first mentioned work [26] had a significantly lower number of participants than the last one [58,59].

### 3.5. Effect of Resveratrol on Obesity

In a crossover study with 11 subjects, RV mimicked the effect of calorie restriction, reducing the metabolic rate, activating AMPK in muscle, and increasing the levels of SIRT1 and peroxisome proliferator-activated receptor gamma coactivator 1 alpha protein. RV also increased the activity of citrate synthase and decreased the lipid content inside the liver, the levels of circulating glucose, triglycerides, alanine aminotransferase, and other inflammation markers. The homeostatic model assessment index was also improved after the intervention [60].

To analyze the longer-term effect of some polyphenols on the metabolic profile, RV and epigallocatechin-3-gallate supplements (80 and 282 mg/day, respectively) were administered during a period of 12 weeks to 38 overweight or obese subjects. An increase in mitochondrial capacity and fat oxidation stimulation was observed, together with a better skeletal muscle oxidative capacity and a preservation of fasting and postprandial fat oxidation. Consequently, triacylglycerol concentration remained unchanged after the RV treatment, unlike in the placebo group, but no improvement of insulin resistance was found in peripheral, hepatic, or adipose tissue [61].

### 3.6. Effect of Resveratrol on Other Health Conditions Associated with Oxidative Stress and Inflammation

The antioxidant effects of RV have been widely studied. In a study with 10 healthy individuals, a single dose of 5 g of RV was given, and a significant increase of tumor necrosis factor-alpha (TNF-α) in plasma was found after 24 h. This enhanced production, as well as the inhibition of IL-10, was confirmed by analysis of peripheral blood mononuclear cells (PBMC), which were activated with different toll-like receptor agonists [62].

In a different trial with nine healthy men and women, who ingested 1 g/day RV capsules for 28 days, RV effect on immune cells was assessed. The results showed that RV induced an increase in circulating T cells and was consequently able to reduce the plasma levels of proinflammatory cytokines TNF-α and monocyte chemoattractant protein 1; moreover, RV significantly increased the plasma antioxidant capacity with a resulting decrease of oxidative stress markers involved in DNA damage [63].

A study consisting in the administration of 75 mg/day of RV per 12 weeks to non-obese, postmenopausal women with normal glucose tolerance, did not observe any change in inflammatory markers, body composition, resting metabolic rate, plasma lipids, liver, skeletal muscle, and adipose tissue volumes, or insulin sensitivity [44].

Lastly, in a study with patients undergoing peritoneal dialysis, the daily consumption of 450 mg of RV over 12 weeks improved urinary ultrafiltration and decreased vascular endothelial growth factor, fetal liver kinase-1, and angiopoietin-2 (angiogenesis markers) when the highest dose was ingested [64].

A possible reason for the contrasting results could be that low doses [44] or a single but higher dose [62] do not have positive health effects and, furthermore, they may cause an acute metabolic stress. On the other hand, a moderate (>450 mg) but continuing intake [63,64] has demonstrated an improved effect of RV. These results suggest that a repeated and moderate administration of RV is better than a single, higher dose administration. Figure 2 schematizes the results of different studies from the last 10 years in which RV had a healthy effect. Additionally, Table 2 presents the details of the effect of RV in each study analyzed. The table is organized according to the participants’ health status.

## 4. Conclusions

In this review, we have described the human clinical trials held in the last decade in which RV was determined in human plasma, urine, or feces. On the one hand, we conclude that the bioavailability and pharmacokinetics of RV depend on the doses ingested, the concomitant ingestion of food matrix, the particle size, the gut microbiota, and the circadian variation.

The results suggest that a repeated and moderate administration of RV is better than the administration of a single, higher dose. A safe and efficient dose is 1 g or more per day; however, RV intake is safe at a dose of up to 5 g, although everyone may experience different adverse effects. Furthermore, the studies showed that RV excretion occurs mainly within the first four hours after ingestion.

On the other hand, RV could have positive effects such as improved antioxidant capacity and modulated neuroinflammation. However, there is disagreement about its positive effects in type 2 diabetes patients and on endothelial function, inflammation, and cardiovascular markers. Contrasting results may be due to the effects of the dose ingested, the gut microbiota status, the health status, and the bioavailability and pharmacokinetics of RV.

It is important to note that the contrasting effects of RV in the different works can be explained by factors such as the number of participants, health status of the gut microbiota, age, gender, lifestyle, dose, administration medium (with or without food), and type of administration (caplet, tablet, powder, gel caps, etc.). For this reason, future studies on RV effects should take into consideration these variables. In addition, further research should be conducted to study in more depth the mechanisms of action of RV in neurological diseases. This review supports the necessity to conduct of larger studies to further investigate the effects of RV on metabolism and neurological functions.

## Figures and Tables

**Figure 1 nutrients-10-01892-f001:**
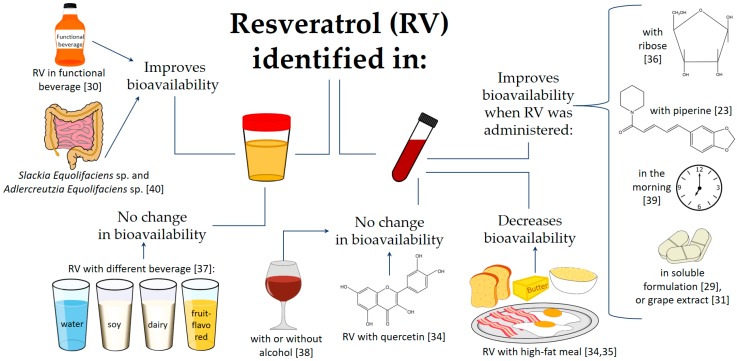
Bioavailability and pharmacokinetics. The image summarizes the results of different studies that identified resveratrol (RV) in urine and blood.

**Figure 2 nutrients-10-01892-f002:**
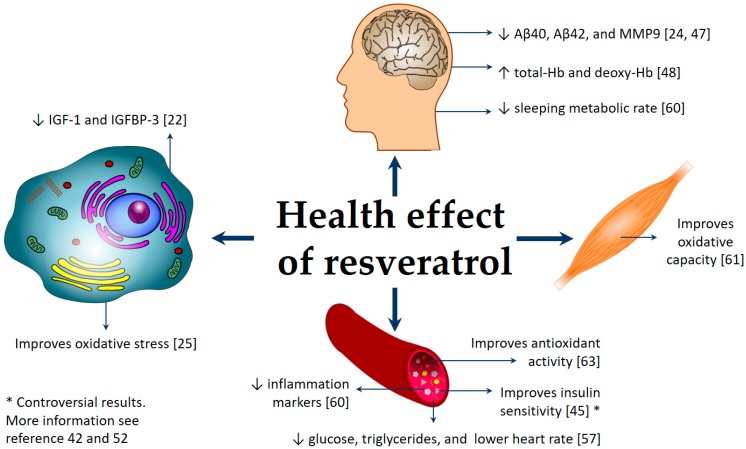
Health effects of resveratrol. A summary of the results of different studies reporting a positive effect of RV on health.

**Table 1 nutrients-10-01892-t001:** Studies of resveratrol bioavailability and pharmacokinetics in the last 10 years *.

Metabolite (form of RV)	Sample	Type of Study	(n)	Dose	Participants’ Health Status	Effect	Ref.
RV and six metabolites: two monosulfates, one disulfate, two monoglucuronides, and one glucuronide-sulfate	Urine and feces	Clinical trial	40	After a five-day washout, 10 subjects received a 0.5 g dose, which was escalated sequentially to 1, 2.5, and 5 g	Healthy	An intake of up to one dose of 5 g of RV was safe, with minor adverse events in some cases; 77% of urinary excretion of RV and its metabolites occurred within four hours after the lowest dose. RV underwent enterohepatic recirculation.	[43]
*t*RV	Plasma	Randomized, crossover, open-label, and single-dose	24	Two treatments with a single dose of 400 mg *t*RV after a high-fat meal or eight hours without breakfast, separated by a washout of seven days or more.	Healthy	The rate of absorption of *t*RV was reduced by the presence of a meal	[35]
*t*RV	Plasma	Phase I, randomized, double-blind, placebo-controlled, and single-center	40	25, 50, 100, or 150 mg, administered at four hours intervals (six times/day) for 48 h (13 doses in total)	Healthy	High daily doses of *t*RV were well tolerated but produced low plasma *t*RV levels; *t*RV bioavailability was higher when it was administered in the morning.	[39]
*t*RV	Plasma	Open-label and single-arm	8	*t*RV 2000 mg twice daily for seven days and *t*RV 2000 mg with quercetin 500 mg twice daily for seven days, with a two-week washout period	Healthy	*t*RV 2000 g twice daily had adequate exposure and was well tolerated by subjects. Moreover, combined intake with quercetin did not influence its exposure	[34]
RV glucuronide and sulfate conjugates, RV glucoside, piceid glucuronides, sulfates, DHR, glucuronide, and sulfate conjugates	Plasma and urine	Randomized and crossover	10	After a three-day washout period, three people were chosen for the pilot study in which they consumed 15 grape extract tablets (total RV 4.72 ± 0.07 mg) with 400 mL of water within 10 min. In parallel, seven people were selected randomly to drink 375 mL of red wine (total RV 6.30 ± 0.09 mg) with 400 mL of water consumed within 10 min.	Healthy	Statistically significant differences between grape extract tablets and red wine treatments were obtained for some metabolites, mainly due to the different composition of RV and piceid from both sources. The grape extract tablets delayed RV absorption compared to the red wine treatment.	[31]
Free RV and conjugated RV (monosulfate, disulfate, and glucoronide)	Plasma	Clinical trial	15	Single dose (40 mg) of *t*RV in soluble formulation or dry powder	Healthy	Bioavailability was higher with soluble formulation compared to dry powder.	[29]
*t*RV, DHR, 3,4′-dihydroxy-*trans*-stilbene, and 3,4′-dihydroxybibenzyl (lunularin).	24 h urine and feces	Controlled intervention	12	Following a washout period, all the subjects received a single oral dose of 0.5 mg *t*RV/kg body weight in the form of a grapevine-shoot supplement (7.7% *t*RV as well as other stilbene mono- and oligomers [14.6%-ε-viniferin, 3.4% ampelopsin A, 1.8% hopeaphenol, 0.6% *trans*-piceatannol, 1.6% r-2-viniferin (vitisin A), 2.5% miyabenol C, 2.5% r-viniferin (vitisin B), and 2.4% iso-*trans*-ε-viniferin].	Healthy	The human gut microbiota produced pronounced interindividual differences in *t*RV. *Slackia Equolifaciens* sp. and *Adlercreutzia Equolifaciens* sp. were identified as DHR producers, but the bacteria that produce dehydroxylated metabolites were not determined.	[40]
*t*RV	Plasma	Pilot study	2	146 +/− 5.5 mg *t*RV per 2000 mg of lozenge mass, containing about 46% ribose, 46% (fructose/sucrose mixture), and 8% *t*RV	Healthy	A mixture of ribose and RV oral transmucosal administration achieved a much higher and quicker RV release compared to the reported traditional free RV capsules.	[36]
Free and conjugated RV	Plasma	Randomized and three-way crossover	15	Oral doses equivalent to 50 mg or 150 mg of *t*RV or plant-derived RV (150 mg) on three occasions separated by seven-day washout periods.	Healthy	150 mg dose of *t*RV showed higher total and free levels than 50 mg dose	[46]
*t*RV, RV, 3-*O*-sulphate, RV 4′-*O*-glucuronide, and RV 3-*O*-glucuronide	Plasma	Randomized, double-blind, and placebo-controlled	23	250 mg of *t*RV or 250 mg of *t*RV with 20 mg of piperine on separate days at least a week apart.	Healthy	Piperine co-supplementation with 250 mg of *t*RV or 250 mg of *t*RV; piperine enhanced the absorption of the polyphenol leading to an increase in cerebral blood flow.	[23]
*t*R4G, *c*R4G, *t*R3G, *c*R3G, *t*R4S, *c*R4S, *t*R3S, *c*R3S, *t*R34dS, RV-SG, *t*piceid, *c*piceid, Pic-G, Pic-S1, Pic-S2, DHR, DHR-G1, DHR-G2, DHR-S1, DHR-S2, and DHR-SG	24 h urine	Randomized, double-blind, placebo-controlled, crossover, and intervention study	26	Consumed twice a day (with breakfast and dinner) for 15 days (per each phase) 187 mL of: a control placebo and a functional beverage (4280 g/L of hydroxycinnamic acids, 16 mg/L of anthocyanins, 96 mg/L of flavanols, 83 mg/L of hydroxybenzoic acids, and 5.7 mg/L of stilbenes)	Healthy	The whole profile of the 21 RV metabolites increased after acute and chronic consumption of the functional beverage with respect to the control-placebo beverage and to the baseline.	[30]
Phenolic acids including, 3-hydroxyphenylacetic acid, 3-hydroxyhippuric acid, 4-hydroxyhippuric acid, and Hippuric Acid,	24 h urine	Randomized, placebo-controlled, and crossover	35	Six placebo gelatin capsules consumed with 200 mL of water (control) Six capsules containing 800 mg polyphenols (141 mg anthocyanins, 24 mg flavan-3-ols, 16 mg procyanidins, 10 mg phenolic acids, 9 mg flavonols, and 1 mg stilbenes) derived from red wine and grape extracts, or the same dose of polyphenols incorporated into one of the following: 200 mL of water (positive control), 200 g of dairy drink, 200 g of soy drink, 200 g fruit-flavored drink, or protein-free drink.	Healthy	Bioavailability of polyphenols and the excretion of their phenolic metabolites were not significantly affected when polyphenols were consumed in protein-rich soy or dairy drinks.	[37]
Total RV	Plasma	Randomized and double-blind	9	5 g/day of SRT501 for approximately 14 days	IV colorectal cancer and hepatic metastasis subjects scheduled to undergo hepatectomy.	RV treatment was well tolerated by the patients. The peak plasma after ingestion of SRT501 was 1.942 ng/m, higher than that of an equivalent dose of non-micronized RV supplementation.	[33]
*t*R4G, *c*R4G, *t*R3G, *c*R3G, *t*R4S, *c*R4S, *t*R3S, *c*R3S, *t*R34dS, RV-SG, *t*piceid, *c*piceid, Pic-G, Pic-S1, Pic-S2, DHR, DHR-G1, DHR-G2, DHR-S1, DHR-S2, and DHR-SG	24 h urine	Randomized, crossover, and controlled clinical trial	59	15-day run-in period in which they consumed neither grape-derived products nor alcoholic beverages. Afterwards, they consumed every day for four weeks: 272 mL of RW (red wine) with 30 g ethanol/day or 272 mL of DRW (dealcoholized red wine), following the same background diet.	High cardiovascular risk	The whole profile of the 21 RV metabolites increased after RW and DRW consumption, and no differences between them were presented	[38]

*c*piceid: *cis*-3,4′,5-trihydroxystilbene-3-β-d-glucopyranoside, *c*R3G: *cis*-RV-3-*O*-glucuronide, *c*R3S: *cis*-RV-3-*O*-sulfate, *c*R4G: *cis*-RV-4′-*O*-glucuronide, *c*R4S: *cis*-RV-4′-*O*-sulfate, DHR: Dihydroresveratrol, DHR-G1: DHR glucuronide 1, DHR-G2: DHR glucuronide 2, DHR-S1: DHR sulfate 1, DHR-S2: DHR sulfate 2, DHR-SG: DHR sulfoglucuronide, Pic-G: piceid-glucuronide, Pic-S1: Piceid sulfate 1, Pic-S2: Piceid sulfate 2, RV: resveratrol, RV-SG: RV sulfoglucuronide, SRT501: microparticular RV of particle size less than 5um, *t*piceid: *trans*-3,4′,5-trihydroxystilbene-3-β-d-glucopyranoside, *t*RV: *trans*-RV, *t*R3G: *trans*-RV-3-*O*-glucuronide, *t*R3S: *trans*-RV-3-*O*-sulfate, *t*R4G: *trans*-RV-4′-*O*-glucuronide, *t*R4S: *trans*-RV-4′-*O*-sulfate, and *t*R34dS: *trans*-RV-3,4′-*O*-disulfate, * Studies which identified resveratrol or some metabolite of resveratrol in plasma, urine, and/or feces.

**Table 2 nutrients-10-01892-t002:** Effects of resveratrol on individuals with different health status reported in the last 10 years *.

Metabolite (form of RV)	Sample	Type of Study	(n)	Dose	Participants’ Health Status	Effect	Ref.
RV 3-*O*-glucuronidated-RV, 4-*O*-glucuronidated-RV, and 3-sulfated-RV	Plasma	Phase II, randomized, double-blind, placebo-controlled, and multi-center	119	500 mg/day RV with 500 mg increments every 13 weeks up to 52 weeks, ending with 1000 mg twice daily	Alzheimer	RV was safe and well tolerated, decreased Aβ40 and MMP9 in CSF, modulated neuroinflammation, and induced adaptive immunity.	[24,47]
RV, RV-3-glucuronide, RV-4′-glucuronide, and RV-3-sulfate RV-4′-sulfate	Plasma	Non-randomized, and open-label	24	Low-dose RV (1 g daily) or high-dose RV (5 g daily) over a 12-week period	Friedreich ataxia	PBMC frataxin protein levels were not affected. High-dose RV treatment showed a beneficial effect on both oxidative stress and some clinical outcome measures.	[25]
Total *t*RV	Plasma	Randomized, double-blind, and placebo-controlled	36	0, 75, 150, and 300 mg at weekly intervals	T2D	A 75 mg dose of RV correlated with an increase in plasma RV concentration, enhanced the cerebrovascular responsiveness to selected stimuli in T2DM adults.	[27]
RV and dehydro RV (aglycones and glucuronide conjugates)	Plasma	Randomized, double-blind, and crossover	17	150 mg/day of resVida (RV) for 30 days	T2D	Intrahepatic lipid content correlated negatively with the plasma RV content. RV plasma levels might be affected by metformin treatment; RV did not improve insulin sensitivity.	[52]
TRM: *t*R3G, *c*R4G, *c*R3G, *t*R4S, *t*R3S, *c*R4S, and cR3S	Urine	Randomized, parallel-group, multi-center, and controlled clinical trial	1000	Exploratory study of the baseline data of PREDIMED study	T2D or at less three major cardiovascular risk factors	Total urinary RV metabolites were directly associated with lower concentrations of fasting blood glucose and triglycerides, and also with lower heart rate. No significant associations were observed between TRM and total cholesterol, HDL, and LDL concentrations, or blood pressure. Therefore, RV may help to decrease cardiovascular risk.	[57]
Total RV	Plasma	Randomized	10	1, 1.5, and 2 g/day RV, taken in divided doses for four weeks	Overweight or obese and insulin resistant.	Fasting glucose was unchanged, but postprandial glucose and three-hour glucose area under the curve decreased significantly. Insulin sensitivity (using the Matsuda index) improved. Fasting lipid profile, CRP, and adiponectin were unchanged.	[45]
Total RV	Plasma	Randomized, double-blind, and placebo-controlled	20	3000 mg/day RV for eight weeks	Overweight or obese with non-alcoholic fatty liver disease	RV did not improve insulin sensitivity, plasma lipids, antioxidant activity, and IGF-1, but it increased ALT and AST, liver enzymes that indicate hepatic stress.	[42]
Total RV	Plasma	Randomized, double-blind, placebo-controlled, and crossover	19	A single dose of RV (30, 90, and 270 mg) administered at one-week intervals over four weeks	Overweight and obese individuals or postmenopausal women with untreated borderline hypertension	Significant linear relationship between RV dose intake and plasma RV concentration. Higher plasma RV concentration was associated with acute flow-mediated dilatation response.	[26]
Total conjugated, unconjugated RV, and DHR	Plasma	Randomized, double-blind, and crossover	11	150 mg/day RV for 30 days	Obese	RV supplementation modestly mimicked the beneficial effects of calorie restriction. It reduced sleeping metabolic rate, affected the AMPK–SIRT1–PGC1α axis, decreased hepatic lipid accumulation. and reduced inflammation markers.	[60]
Epigallocatechin-3-gallate, RV, and DHR	Plasma	Randomized, double-blind, placebo-controlled, and parallel intervention	38	Epigallocatechin-3-gallate + RV 282 and 80 mg/day, respectively for 12 weeks	Overweight and obese	The supplementation improved skeletal muscle oxidative capacity, preserved fasting and postprandial fat oxidation, and prevented an increase in triacylglycerol concentrations.	[61]
Total RV and DHR (both free and conjugated)	Plasma	Randomized, placebo-controlled, and crossover	45	150 mg/day RV capsule for four weeks, with a four-week wash-out period	Overweight and slightly obese	RV did not have an effect on cardiovascular risk metabolic markers, endothelial function, or inflammation.	[58,59]
Total RV and DHR (free and conjugated forms)	Plasma	Randomized, double-blind, and placebo-controlled	45	75 mg/day (99% pure *t*RV), for 12 weeks.	Lean and overweight, postmenopausal	RV supplementation did not change plasma substrates and hormones (glucose, plasma lipids, and insulin), adiponectin, leptin, CRP, and IL-6.	[44]
Total RV, RV glucuronide, and RV sulfate	Plasma	Randomized, double-blind, placebo-controlled, and crossover	22	250 and 500 mg *t*RV on separate days. On three visits, the participants received two single-dose capsules. The capsules were combined to give the following treatments: 1) inert placebo, 2) 250 mg *t*RV, and 3) 500 mg *t*RV.	Healthy	RV intake increased total-Hb and deoxy-Hb concentration, variables related to cerebral blood flow.	[48]
*t*R4G, *t*RDS, *t*R3G, *t*R4S, and *t*R3S	Plasma	Pilot study, randomized, open-label, single-dose, and parallel-group	10	Single 5 g dose	Healthy	RV increased TNF-α level 24 h after supplementation, by an average of 3.5 pg/mL, compared with placebo. High levels of sulfo- and glucuronide-conjugated RV compounds.	[62]
RV	Plasma	Phase I and randomized	9	1000 mg/day RV for 28 days	Healthy	RV was associated with an increase in the number of circulating γδ T cells and regulatory T cells and higher plasma antioxidant activity.	[63]
RV-3-*O*-sulfate, RV-4′-*O*-glucuronide, RV-3-*O*-glucuronide,	Plasma	Clinical trial	40	0.5, 1.0, 2.5, or 5.0 g/day RV for 29 days	Healthy	Treatment with 2.5 g RV decreased IGF-1 and IGFBP-3 levels in all volunteers; RV might contribute to cancer chemoprevention.	[22]

Aβ40: beta amyloid 40, ALT: alanine aminotransferase, AMPK: adenosine monophosphate-activated protein kinase, AST: aspartate aminotransferase, CRP: C reactive protein, *c*R3G: *cis*-RV-3-*O*-glucuronide, *c*R3S: *cis*-RV-3-*O*-sulfate, *c*R4G: *cis*-RV-4′-*O*-glucuronide, *c*R4S: *cis*-RV-4′-*O*-sulfate, CSF: cerebrospinal fluid, deoxy-Hb: deoxygenated Hb, DHR: Dihydroresveratrol, Hb: hemoglobin, IGF-1: insulin-like growth factor 1, IGFBP-3: insulin-like growth factor binding protein 3, IL-6: interleukin-6, MMP9: matrix metalloproteinase 9, PBMC: peripheral blood mononuclear cells, PGC1α: peroxisome proliferator-activated receptor gamma coactivator 1 alpha, RV: resveratrol, SIRT1: sirtuin 1, T2D: type 2 diabetes, TNF-α: tumour necrosis factor alpha, TRM: total RV metabolites, *t*R3G: *trans*-RV-3-*O*-glucuronide, *t*R3S: *trans*-RV-3-*O*-sulfate, *t*R4G: *trans*-RV-4′-*O*-glucuronide, *t*R4S: *trans*-RV-4′-*O*-sulfate, *t*RV: *trans*-RV, and *t*RDS: *trans*-RV-disulfate. * Studies which identified resveratrol or some metabolite of resveratrol in plasma, urine, and/or feces.

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
