# Peer review of "Health Effects of Resveratrol: Results from Human Intervention Trials"

_nutrients, 2018, doi:10.3390/nu10121892_

Reviewer 1 Report

With this review, the authors summarize results so far of clinical trials using resveratrol supplementation carried out in the last decade. They describe its bioavailability and pharmacokinetics, and its effects on a variety of diseases. They provide an exhaustive list of studies carried out so far, which point towards potentially beneficial effects of this polyphenol, warranting future research. However, this review is lacking a few points (described below), in order to improve its understanding and content. A critical analysis or insights from the authors themselves would also greatly improve this manuscript. Please consider incorporating some of the following minor points: English revision: The text needs revising on a whole to improve fluidity and understanding. Please consider some subheadings in section 2 to improve structure also. Style errors/typos: Examples (but not exhaustive list) include: * Lines 80, 81, 83, 91: There should always be a space between a number and the symbols “h”, ng/mL”, etc. These are just some examples, please revise all text. * Generally, numbers from 1->9 are written (i.e. one, two, three...). Please consider revising. * “ml” vs. “mL”, “h” vs. “hours”, etc. Please standardize. * Table 1. First page, last study. There should be no hyphen between “15” and “grape” * At the end of a list, a comma is added before “and” in USA English. This is not so in British English. Please standardize throughout text, since both forms have been used. * Figure 1: “high fat” should be “high-fat”, as per written in the text. * Line 169: “Twenty” should be “20” Abstract: * Please consider revising abstract on a whole. For example, the first sentence of the abstract is slightly confusing and could be improved. Introduction: * Considering this is a revision on the effects of resveratrol on human health, a brief explanation of current state of affairs of its supplementation would greatly improve this section. Is there a recommended dose? What is the current consumption? Is it being used to treat diseases? Etc… * The description of the aim could be improved. Bioavailability and pharmacokinetics of resveratrol * A brief explanation (short sentence) on what phase I and II metabolism are would aid the reader and introduce the topic. Table 2: Table 2 is an exhaustive and interesting list of studies showing an effect of resveratrol on particular diseases. However, it is somewhat hard to immediately pick out which diseases it affects or in which order these studies have been presented. It would be interesting for the authors to consider presenting this table in a more understandable format so that the reader can easily pick out the diseases mentioned (for example) or give an idea of why these studies have been presented in this particular order. Effect of resveratrol: Please consider improving this title. Effect of resveratrol on…. * A brief introduction before the first subheading of this section would improve overall understanding. * Line 191: Some studies have analyzed the effect of RV on obesity. This sentence is not very informative. Why have some studies focused on obesity? What is their basis? Some background information (brief) would help the reader follow the text. Conclusion: The authors provide a quick summary of the results, which although it is interesting, it would really add to the review to expand this further (either in another section previous to conclusions or in conclusions itself). Examples: What are the main pitfalls of studies so far? How can they be improved in future? What is the future for resveratrol in human health? Why should more studies be carried out? Should particular forms be considered over others in terms of bioavailability? Supplementation vs. diet…? This could then tie in with the introduction.

Author Response

Thank you for your comments. We attached our revised version.

Reviewer 2 Report

Ramírez-Garza et al. aim to summarize the available data derived from clinical studies on resveratrol’s bioavailability and a variety of clinical endpoints. As the authors state correctly, there is a plethora of very heterogeneous clinical evidence in this field and a summary is therefore highly welcome. The manuscript is well written and the graphical support is appealing and clear. However, I have some comments, which need to be addressed before publication. Major points: 1) The section summarizing the data on bioavailability is very clear and straight-forward. However, in the sections summarizing the clinical effects, very heterogeneous studies are reported without any conceivable connection, and the headlines do not really apply to the reported studies. In the metabolic sections there are reports on dialysis studies (Lin et al.) as well as immunological studies (Gualdoni et al.), which do not seem to relate to any metabolic disorders (the study by Gualdoni et al. should also be referenced in the bioavailability section). There are reports on Friedreich ataxia in the metabolic section, why is this not discussed in the neurological section? The reports on metabolic alterations are interrupted by reports on cardiovascular effects. Please re-order the sections in a manner that facilitates following the respective thoughts. 2) Although not a major objective in a review of this kind, there needs to be a more thorough assessment of the study quality when reporting the respective findings. For instance, the study of Lin et al. which is carelessly discussed to show “positive effect on urinary ultrafiltration and improved angiogenesis“ has major limitations concerning randomization, statistical analysis and a potential attrition bias, and the data does not justify the statement that there is (some angiogenesis markers were measured, not angiogenesis). This also applies to other studies reported. The authors need to better balance their reporting of these studies. Minor points: 1) Line 244&245, “RV clearly has an important antioxidant effect, and modulates neuroinflammation and cancer“: You have not discussed evidence justifying this statement, please modify

Author Response

(The authors gave the same response as above.)

Reviewer 3 Report

The review by Ramirez-Garza et al summarizes the results of different clinical trials conducted over the last decade, with a particular focus on bioavailability and metabolism rather than on clinical efficacy. The review is clear and well written, though too synthetic. The authors should have discussed in more details the studies and try to explain why some data are conflicting.

A few examples are:

-the references 46 and 47 regarding the levels of plasma TNF-alpha after RV treatment indicate conflicting results. Is the modulation of cytokine plasma levels related to the lenght of treatment?

-in the cognitive performance paragraph, one trial (ref 24) with RV seems to reduce brain volume. This effect would merit a comment, and should be compared with that of other known anti-alzheimer drugs.

Minor point: a few mispellings are present in the paper:

-line 78: Pharmacokinetic studies reveal, not reveals

-line 136: The gut microbiota plays, not play

-line 170-72: please specify that the TNF-alpha alteration was observed at the plasma level

-fig 2: please specify that in the cerebrospinal fluid is the Abeta40 that is lower, not the fluid itself.

Author Response

(The authors gave the same response as above.)
